# Investigation of Weather Triggers Preceding Outbreaks of Acute Bovine Liver Disease in Australia

**DOI:** 10.3390/toxins14060414

**Published:** 2022-06-17

**Authors:** Eve M. Manthorpe, Grant T. Rawlin, Mark A. Stevenson, Lucy Woolford, Charles G. B. Caraguel

**Affiliations:** 1School of Animal and Veterinary Sciences, The University of Adelaide, Adelaide, SA 5371, Australia; lucy.woolford@adelaide.edu.au (L.W.); charles.caraguel@adelaide.edu.au (C.G.B.C.); 2Department of Jobs, Precincts and Regions, Agribio, The Centre for AgriBioscience, Melbourne, VIC 3083, Australia; grant.rawlin@agriculture.vic.gov.au; 3Melbourne Veterinary School, Faculty of Veterinary and Agricultural Sciences, The University of Melbourne, Melbourne, VIC 3010, Australia; mark.stevenson1@unimelb.edu.au

**Keywords:** acute bovine liver disease, cattle, liver, mycotoxin, plant toxin, weather

## Abstract

Acute bovine liver disease (ABLD) is a hepatic disease affecting cattle sporadically in southern Australia, characterised histologically by striking periportal hepatocellular necrosis. The cause of ABLD is unknown; however, the seasonality and acute presentation of outbreaks suggest mycotoxin involvement. We described the geographical and seasonal occurrence of ABLD reports from 2010 to 2020 in Victoria, Australia, and explored potential weather triggers preceding 26 outbreaks occurring across 23 properties using a case-crossover design. Outbreaks occurred most frequently in autumn/early winter and in herds located along the southern coastal plain of Victoria, and occasionally within the low-lying regions of the Great Dividing Range. Lactating adult dairy cattle represented the most reported cases. We observed a significant association between an increase in average daily dewpoint in the 15 days preceding an ABLD outbreak, suggesting that dew formation may be a key determinant for this disease. Our findings support the etiology of a potent hepatotoxic agent that requires moisture for proliferation and/or toxin production.

## 1. Introduction

Acute bovine liver disease (ABLD) occurs sporadically throughout the south-eastern regions of Australia without a confirmed etiology [1]. Cattle in the states of Victoria, Tasmania, and occasionally in the south-eastern parts of South Australia are most affected. Affected cattle show non-specific clinical signs of acute hepatic disease, with marked elevations in hepatocellular enzyme activities [2]. Clinical progression is often rapid, with death or euthanasia occurring within 48 h of the onset of clinical signs [2]. Histopathologic features include striking acute periportal necrosis, which extends to massive (panlobular) necrosis in a large proportion of cases [2]. Histologic differential diagnoses are therefore limited to other causes of periportal to massive hepatocellular necrosis affecting cattle in these regions [2]. These include Cyanophyceae spp. (*Anabena* spp. and *Microcystis* spp.) [1], Myoporaceae spp. [3], amatoxin (*Amanita* sp., *Galerina* sp., *Lepiota* sp.) [4], and copper [5], usually from parenteral administration [2,6,7]. Thus, a diagnosis of ABLD relies on histologic assessment and thorough-onsite examination.

The cause of ABLD is currently unknown; however, the seasonality and histologic lesions are most suggestive of mycotoxin involvement [2]. Notably, all previous case reports are restricted to cooler and higher rainfall areas of southern Australia [2,8,9,10]. Previous case reports suggest an increased occurrence in cattle grazing lush pasture species that have abundant decaying litter from senescent previous growth, particularly in autumn [1,2,8,11]. As such, the apparent seasonality of disease may be linked to weather conditions that favour rapid pasture growth [8]. Other known toxins have been excluded through thorough onsite examination, resulting in a focus on previously unknown toxic agents [9,10]. Outbreaks have been consistently reported with the presence of dry, senescent *Cynosurus echinatus* (rough dog’s tail grass); however, in vivo trials have not demonstrated toxicity of the grass itself [1,11]. Of the outbreaks investigated for fungi, *Pyrenophora* spp. (previously *Dreschlera* spp.) have been detected at all outbreaks. Studies have demonstrated cytotoxic effects of the fungal genus on rat hepatocytes in vitro [1], but this was not repeatable in vivo [11]. The significance of *C. echinatus* and *Pyrenophora* spp. in events of ABLD is unclear, and may be limited to an indication of sub-optimally managed pasture, which is likely to contain many weeds and potential toxic substrates [1]. It is now thought that any putative mycotoxin may only be produced at specific stages in the lifecycle of the fungus, such as when actively sporulating, a process that requires specific weather conditions [1,11].

The described climatic conditions conform to those preceding other mycotoxic hepatopathies, supporting current suspicions of mycotoxin involvement, and may be an indication of the presence of a suitable litter substrate and microclimate for fungal growth [3,11]. Investigations involving *Pyrenophora* spp. and *C. echinatus* as sufficient causes of disease have been unsuccessful [11], indicating a need to widen investigations. Despite the various environmental and climatic observations, studies characterising distinct weather conditions preceding ABLD outbreaks are not present in the literature. Here, we systematically investigated the measurable weather parameters in the 15 days preceding 26 natural ABLD outbreaks across Victoria, Australia. We aimed to elucidate any significant weather indicators precipitating or preceding disease, as well as any evidence to support mycotoxin involvement in the pathogenesis of disease, to guide further epidemiologic and etiologic investigations.

## 2. Materials and Methods

### 2.1. Study Design

We investigated potential associations between weather predisposing factors and ABLD case occurrence using a unidirectional time-stratified case-crossover study design. The weather conditions in the 30 days preceding an ABLD case (index case window) were compared with the weather conditions at the same location and the same calendar period for the two previous calendar years (two matching referent windows) when no cases were reported. The matching of one index weather window with two referent windows allowed to control for potential time-invariant confounders. Seasonality and long-term effects were also controlled for by matching the windows based on the same calendar period up to two years prior. However, unmeasured confounder factors varying year-to-year may not have been controlled with this design.

### 2.2. Data

#### 2.2.1. Acute Bovine Liver Disease Cases

Standard records from 26 ABLD outbreaks, involving one or more animals, were accessed from diagnostic reports from the Centre for AgriBioscience (Melbourne, VIC, Australia), between the 1st of January 2010 and the 31st of December 2020. Some properties experienced multiple outbreaks during a similar time frame (Table 1 and Appendix A). These were considered separate events if they commenced at distinct dates in different herds (property J), or at different geographic locations owned by the same producer (property E). Only records from acute hepatotoxicosis consistent with ABLD diagnosed by a veterinary pathologist were selected. These included cases diagnosed by either histopathological examination of liver specimens collected at post-mortem, or biochemical evaluation and exclusion of other hepatotoxic agents through paddock and water examination. The case date began at the time of clinical onset (time of initial onset of clinical signs as reported by the producer), or, when this was not available, the date of specimen collection by the attending veterinarian. The available data from case reports included property identification and location, the date of specimen collection, the signalment and number of affected animals, clinical history and presentation, the total number of animals at risk, and the estimated financial loss to the producer. Clinical history and presentation were variably provided, but often included the date of clinical onset, clinical signs seen in affected animals, post-mortem findings, diet of affected animals, lactation or gestation status, and the dates of stock movement.

#### 2.2.2. Weather Data

Australia-wide daily gridded weather data was sourced from the Australian Bureau of Meteorology (BoM), including minimum and maximum temperature (°C), daily solar exposure (MJ/m^2^), total daily precipitation (mm), and vapour pressure (hPa) over the 2008–2020 calendar year period to cover back to two years of all reported cases’ dates (Appendix A). Data were provided as a set of daily raster maps with a 5 km grid for each weather parameter. For each index and matching referent windows, arrays of weather parameter values were extracted into separate databases for the longitude and latitude of each case location.

### 2.3. Statistical Analysis

#### 2.3.1. Weather Data Handling and Summary

For each case, we derived the daily average temperature (mean) and the daily temperature gap (difference) from the daily minimum and maximum temperatures. The daily dew point temperature (*T_d_*) at 9 am and 3 pm were calculated from the respective vapour pressure (*P_v_*) using formulae derived from BoM [12]:(1)$$T_{d}\left({}^{\circ}C\right)=\frac{-237.31\ln\left(P_{v}\right)+429.41808}{\ln\left(P_{v}\right)-19.079025}$$

The relative humidity (*U*) was then calculated from the *T_d_* and minimum temperature (*T*) as follows [13]:(2)$$U=100*\;\frac{e\left(1.8096+\left(\frac{17.2694T_{d}}{237.3+T_{d}}\right)\right)}{e\left(1.8096+\left(\frac{17.2694T}{237.3+T}\right)\right)}$$

Five-day moving averages across the 26 index or referent windows were superimposed and graphed for each weather parameter for up to 30-days. Visual exploration of the index and referent trends guided the decision for the final window size of interest.

#### 2.3.2. Data Analysis

We summarised the weather parameters at each case location by averaging daily values over the refined window periods and, for daily precipitation and solar exposure, we cumulated these values over these periods (Appendix A). Summary weather data values of index windows were matched to two referent window values for each ABLD event and then analysed as matched case-control data using conditional logistic regression. The regression was first run using one single predictor at a time (univariable analysis). Multivariable analysis was considered only for weather parameters strongly associated with the occurrence of an ABLD event after exploring potential strong collinearity with other significant weather parameters (Appendix A). Significant comparisons were interpreted at the 5% level. The analysis was implemented in the statistical package Stata v.17.1 (StatCorp Ltd., College Station, TX, USA).

The strength of association between prior weather conditions and an ABLD event were estimated using conditional odds ratios (OR). OR is a symmetrical measure and, regardless of the directionality of the case-control analysis, it can be interpreted as the comparison of the odds of an event occurring in an exposed group, compared to the odds of the same event occurring in a non-exposed group. In the current study, “exposed” referred to exposure to certain weather conditions summarised over a period of time and measured on a continuous scale. An OR above one indicates that, for each unit increment of the weather summary, the odds of ABLD increases, while an OR below one indicates a decrease in ABLD odds. An OR non-significantly different from ‘one’ (i.e., 95% CI includes one) indicates no association between the weather summary of interest and ABLD.

The cessible series of 26 outbreaks, matched with two referent windows each, provided sufficient study power (1-type-II-error = 80%) to detect significantly (type-I-error = 5%) an odds ratio of at least 3.9 assuming a binary exposure (calculated using Stata command ‘power mcc’). This confirms that only associations of strong magnitude (OR ≥ 2 with our continuous data) could be significantly detected with the available records.

## 3. Results

### 3.1. Case Description

A total of 26 acute bovine liver disease (ABLD) outbreaks were reported across 23 properties in Victoria from the 1st of January 2010 to the 31st of December 2020. A descriptive summary of the outbreaks is presented in Table 1. A presumptive diagnosis was made by histologic examination of liver specimens in 21 cases, and examination of biochemistry data in five outbreaks, combined with historical and environmental exclusion of other potential aetiologic agents (Table 1). Environmental examination of paddocks and water sources ruled out Myoporaceae spp., Cyanophyceae spp., copper, or amatoxin involvement in all cases (Table 1). Most events occurred along the coastal plain of Victoria (Figure 1), south of the Great Dividing Range (GDR), within 65 km of the coastline. This is a cool and temperate region, with the mildest climate of all Victorian regions. Summers are mild to warm, with the hottest temperatures in January and February, and winters are mild to cool, coldest in July. Rainfall is greatest during winter, although early summer rainfall may be heavy. Sixteen events occurred west of Melbourne, between Geelong and Mount Gambier. The remaining 10 were scattered throughout eastern Victoria, within and around the Great Dividing Range (GDR). One event (no. 24) occurred 170 km from the coastline, situated within the north-west border of the GDR. The altitude of the Australian Alps ranges from just a few hundred metres above sea level to the top of Mt Kosciuszko at 2228 m. They experience a mid-latitude mountain climate, with no dry season and a mild summer. Precipitation occurs all year round, most often in spring and winter, when most precipitation falls as snow. The environmental lapse rate indicates that for every 1000-m rise in altitude there is a 6.5 °C drop in ambient air temperature. Event 24 occurred at 281 m above sea level, indicating milder general temperatures than surrounding high-lying regions. The coastal plain and Alpine regions are generally wetter than non-coastal, low-lying parts of Victoria, except for areas to the west of Melbourne, where annual rainfall may drop below 600 mm. Land use in the southern coastal plain is dominated by primary production and conservation reserves.

Of the 26 outbreaks reported, twenty-one (81%) outbreaks occurred in autumn, particularly in the month of April (15/26 (58%)), three (11%) occurred in winter, and two (8%) occurred in Spring. Fifteen (58%) of those outbreaks occurred in mid-lactation dairy cows, one (4%) occurred in dry cows, one (4%) occurred in a male Angus, one (4%) occurred two to three days post-weaning, seven (27%) outbreaks occurred in Holstein or Holstein-cross cows at an unspecified stage of lactation, and one (4%) occurred in an Angus cow at an unspecified stage of lactation. Outbreaks were most common in adult cows (19/26 (73%)). Ages in six (23%) outbreaks were unspecified, and the remaining outbreak (1/26 (4%)) occurred in heifers. Morbidity and mortality rates ranged from 1% to 65%, and 0% to 62% respectively.

### 3.2. Weather Predictors

Visual comparison of weather trends in index and referent windows led to refinement of the window size to 15 days (Figure 2 and Figure 3). Weather summaries over these periods confirmed collinearity with interdependent weather variables (Appendix A). Average dew point temperatures showed a significant positive correlation with vapour pressure, and relative humidity showed a significant negative correlation with average air temperature. Additionally, dew point temperature was significantly and positively correlated with air temperature, demonstrating that as air temperature increases, so too does the dew point temperature. Dew point temperature was significantly and negatively correlated with relative humidity. This is expected when evaporation is considered: as the dew point temperature decreases, the minimum temperature required for dew formation also decreases, reducing the chance of condensation on the ground (dew) and means that the water content of air (relative humidity) is increased.

Estimates of unifactorial associations between weather parameters and an ABLD event are reported in Table 2. Average daily dew points, both at 9 am and 3 pm, were significantly associated with ABLD occurrence (odds ratio (OR) = 2.01, 95% confidence interval (CI): 1.20–3.37; odds ratio (OR) = 1.83, 95% confidence interval (CI): 1.15–2.90, respectively), i.e., for every 1 degree Celsius (°C) increase in the average dew point in the preceding 15 days, the odds of an ABLD event approximately doubled. Additionally, a significant negative association with ABLD outbreaks was seen for the average gap between daily maximum temperature and dew point at 3 pm. When the average gap increased by 1 degree Celsius (°C) in the 15 preceding days, the odds of ABLD decreased by 27% (odds ratio (OR) = 0.73, 95% confidence interval (CI): 0.54–1.00). Graphed data visually shows the increase in average dew point, and the decrease in dew point and air temperature gap, in cases compared with referent windows (Figure 2 and Figure 3). A higher dew point means a lower minimum temperature requirement for dew formation, while a lower gap indicates an increased likelihood of dew formation; however, this was only marginally significant when considering 3 pm dew point and maximum air temperature (*p*-value = 0.047).

## 4. Discussion

Outbreaks of ABLD are reported inconsistently along the Victorian coastal zone, making prospective investigations difficult. Access to retrospective records of 26 events over an 11-year period provided an opportunity to produce preliminary insights into the ecology of this disease, with hopes of guiding more targeted investigations. There were several study constraints, many of which are common to retrospective investigations. Most importantly, outbreak records were limited and did not enable robust causal inferences. Some of the weaker associations observed in this study require further evaluation with larger sample sizes to determine their significance. The lack of confirmed non-case properties limited study design control options. Reliance on diagnostic submissions made it impossible to confirm the complete absence of ABLD during the selected control years, leaving the possibility that ABLD occurred at a level that did not warrant reporting or diagnostic intervention. The implementation of a case-crossover design enabled us to investigate measured site-specific factors that varied between index and referent windows, such as weather parameters. However, this design would fail to identify unconsidered confounding factors that also vary from year to year within a property (e.g., biotic factors or husbandry practices). The putative fungal etiology of ABLD indicates the likely interplay of weather variables and potential toxic substrates in this disease. Ideally, on-farm investigations of ABLD would include prospective comparison of weather data with in-depth identification of all potential plant sources of ABLD-toxin on the case properties studied, and the inclusion of controls which have been confirmed to be unaffected by ABLD. A further drawback in the study of ABLD is the current diagnostic criteria, which are limited to histopathological evidence of periportal to massive hepatocellular necrosis, or in cases where this is not available, biochemical evidence of significant hepatocellular insult, combined with environmental exclusion of other potential causes. This of course increases the risk of false case selections that have not been confirmed as ABLD. Unfortunately, this cannot be avoided until an etiologic agent is found.

The regional distribution of ABLD outbreaks has long roused suspicion for a key role of weather or other environmental factors in the development of disease. The current study provides an initial insight into the specific weather variables involved in ABLD occurrence and provides further support for the described geographic distribution, signalment, and seasonality of disease. Notably, we found a significant positive association between the occurrence of ABLD and the dew point. Dew point is the temperature to which air must be cooled to produce condensation (dew). Both dew point and relative humidity reflect air moisture; however, relative humidity, which expresses the relative saturation of the air, is strongly influenced by temperature—warm air can hold more moisture, resulting in a negative correlation between air temperature and relative humidity. In contrast, dew point is related to the quantity of moisture and is more robust when compared with relative humidity, but is dependent on, and positively correlated with vapour pressure [14]. Our results indicate that an increase in average dew point by 1 °C across 15 days increases the risk of an outbreak by a factor of 2.01 (i.e., doubles the risk of an outbreak). Additionally, a significant negative association was found in the average gap between dew point at 3 pm and maximum temperature, with a 27% decrease in outbreak risk for every 1 °C widening in the gap over a 15-day period. Although this is more complicated to replicate in vivo, this association suggests that when the dew point and maximum temperature are further distanced, and dew is less likely to persist later in the day; as such, the risk of an outbreak is decreased. Therefore, dew formation appears to be a key indicator for disease.

The apparent requirement for dew formation in the development of disease is highly suggestive of a mycotoxicosis. Mycotoxins are compounds produced by various species of fungi, which can grow on feed, and cause significant disease in plants, animals, and humans [15]. Although there are many factors involved in mycotoxin synthesis, climate is the most important factor, with several weather variables influencing production and dispersal of the agent [15,16]. Notably, water content is a limiting determinant for fungal growth and mycotoxin production. This is measured under laboratory conditions by water activity (a_w_), a ratio between the vapor pressure of the growth media itself, and the vapor pressure of distilled water under identical conditions: a_w_ of 0.80 means the vapor pressure is 80 percent of that of pure water. Many mycotoxins have a defined minimum and optimal a_w_ that is required for production [15]. For example, the minimum a_w_ for ochratoxin A formation is 0.80 a_w_, with optimum production at 0.96–0.98 a_w_, and the minimum a_w_ for patulin production is 0.82–0.83 [15,17]. It is therefore reasonable that any measure of moisture availability will have similar limiting or potentiating effects on mycotoxin production. Where dew formation has been specifically investigated, it too has been shown to be associated with fungal growth and/or mycotoxin production: dew during warm periods is associated with increased crop aflatoxin levels, and heavy dew is highly favourable for growth of various *Fusarium* spp. responsible for “Fusarium head blight”, a disease affecting wheat and barley crops [15,16]. In this case, dew formation is the first environmental evidence of mycotoxin involvement in events of ABLD.

The increased risk of ABLD outbreaks in association with dew formation should be clinically applied with consideration of the season, property location, signalment of at-risk animals, and pasture management practices. Regarding geographic distribution, it appears properties within the southern coastal plain of Victoria, or, less commonly within low-lying regions of the GDR, are at an increased risk of ABLD. Additionally, ABLD occurred primarily in autumn. The reason for the geographic and seasonal clustering of ABLD occurrence is yet to be fully elucidated but is likely attributable to some interplay between the presence of appropriate litter substrate for toxin production, and the occurrence of specific weather conditions, such as dew. For example, although dew is common in spring, the required plant material may not be present; namely, senescent plant material that is typically abundant during autumn. Other weather variables that may explain the risk of disease occurrence in autumn could not be discerned in the current study, and marginal associations may have been overlooked due to the low study numbers. A key benefit of a dew point contribution is the relative ease with which this can be assessed by veterinarians and producers. High-risk properties may therefore implement management strategies if the frequency of dew begins to increase.

Previous publications have indicated that disease is more common in lactating dairy cattle [2,11]. The current study population was likewise predominated dairy cows in mid-lactation (15/26 (58%)), with 7/26 (27%) events occurring in cows with an unreported lactation status, and only sporadic reports in dry cows and beef breeds. However, there are several explanations beyond an intrinsic susceptibility to disease: first, the climatic conditions in these regions are favourable for dairy production, but equally may provide the required climatic conditions for outbreaks; second, dairy cattle in these regions tend to be rotationally grazed (or strip grazed), increasing the risk of access to a potential toxic agent; finally, the high energy requirements of lactation may reduce grazing discrimination, increasing the risk of unpalatable toxin ingestion.

Properties should be considered at higher risk for ABLD if they fulfill a combination of the following criteria: herds in mid-lactation, rotational grazing, pasture with senescent material and lush new growth, previous outbreaks of ABLD, and autumn to early winter season. Management strategies may include grazing the paddock with sheep before introducing cattle, cultivating high-risk paddocks, avoiding the use of paddocks with abundant dry material, and using a few animals to ‘test’ for toxicity on previously toxic paddocks [18].

## 5. Conclusions

We provide an initial insight into the ecology of ABLD, with dew point likely playing a determinant role in the etiopathogenesis of the disease. Notably, we provide further evidence for the involvement of a harmful agent that requires moisture for proliferation and/or toxin production. More work is required to determine the potential interplay of weather and pasture types with hopes of determining an etiologic agent(s).

## Figures and Tables

**Figure 1 toxins-14-00414-f001:**
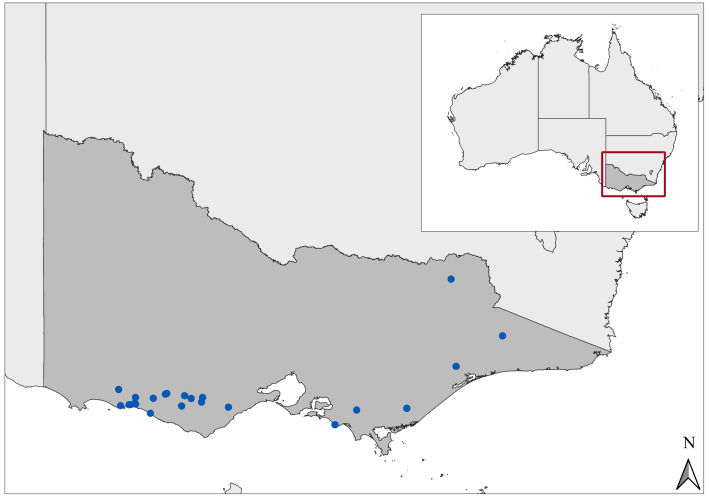
Geographic distribution of 26 outbreaks of acute bovine liver disease on 23 properties in Victoria, Australia, 2008–2020.

**Figure 2 toxins-14-00414-f002:**
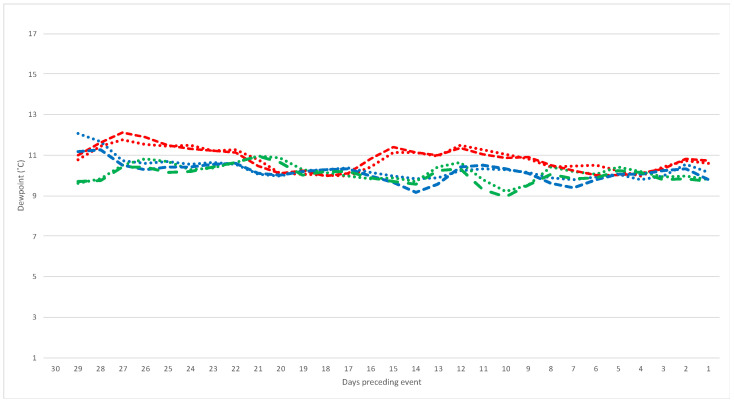
Five-day moving average of dewpoint at 9 am (dots) and 3 pm (dash) for the 30 days preceding 26 events of acute bovine liver disease in an outbreak year (red), and the same calendar window of the preceding 2 years (green, year 1; blue, year 2). Fainted lines denote the raw values for each event to visualise actual data variability.

**Figure 3 toxins-14-00414-f003:**
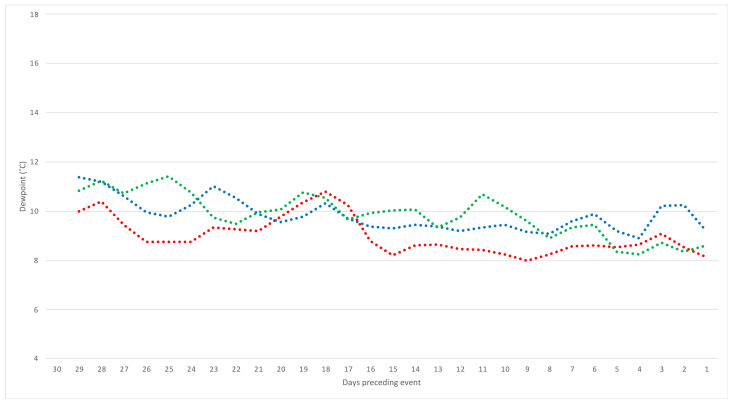
Two-day moving average of maximum temperature and 3 pm dewpoint temperature gap for the 30 days preceding 26 events of acute bovine liver disease in an outbreak year (red), and the same calendar window of the preceding 2 years (green, year 1; blue, year 2). Fainted lines denote the raw values for each event to visualise actual data variability.

**Table 1 toxins-14-00414-t001:** Morbidity and mortality rates for outbreaks of acute bovine liver disease, including property and case identification, date of outbreak, and signalment of affected animals.

Property ID	Outbreak ID	Outbreak Date	Season	Morbidity (%, n)	Mortality (%, n)	Signalment	Lactation Status	Diagnostic Method (Number of Individuals Examined)
Property A	1	April 2008	Autumn	10 (15/150)	4 (6/150)	F, A, Holstein	n/a	H (1)
Property B	2	May 2008	Autumn	10 (30/300)	1 (3/300)	F, A, Holstein	Mid-lactation	H (2)
Property C	3	April 2010	Autumn	22 (11/50)	2 (1/50)	F, n/a, n/a	n/a	B, E (1)
Property D	4	April 2010	Autumn	n/a	n/a	F, n/a, n/a	n/a	H (1)
Property E	5	April 2010	Autumn	25 (50/200)	1 (3/200)	F, A, Holstein	Mid-lactation	H (10)
Property E	6	April 2010	Autumn	15 (30/200)	2 (5/200)	F, A, Holstein	Mid-lactation	H (1)
Property F	7	April 2010	Autumn	18 (50/270)	1 (3/270)	F, A, Holstein	Mid-lactation	H (6)
Property G	8	March 2011	Autumn	10 (20/200)	1 (2/200)	F, A, Holstein	Mid-lactation	H (2)
Property H	9	April 2011	Autumn	5 (10/200)	0.5 (1/200)	F, 16 m, Holstein	Mid-lactation	H (1)
Property I	10	May 2013	Autumn	1 (6/400)	1 (6/400)	F, A, Holstein	Mid-lactation	H (4)
Property J	11	June 2013	Winter	53 (80/150)	15 (22/150)	F, A, Holstein	Mid-lactation	H (3)
Property J	12	July 2013	Autumn	4 (4/100)	1 (1/100)	F, A, Holstein	Mid-lactation	H (3)
Property K	13	April 2014	Autumn	n/a	62 (20/32)	F, n/a, Angus	2–3 d post-weaning	H (1)
Property L	14	April 2014	Autumn	6 (15/250)	1 (3/250)	F, A, Holstein	Mid-lactation	H (3)
Property M	15	April 2014	Autumn	15 (15/100)	0 (0/100)	F, A, Angus	Dry	B, E (1)
Property N	16	April 2014	Autumn	5 (7/150)	0 (0/150)	F, A, Holstein	Mid-lactation	H (5)
Property O	17	April 2015	Autumn	33 (10/30)	13 (4/30)	F, A, Angus	n/a	H (4)
Property P	18	November 2017	Spring	65 (150/230)	12 (27/230)	F, A, Holstein	Mid-lactation	H (2)
Property Q	19	April 2018	Autumn	9 (15/160)	4 (7/160)	F, A, Holstein	Mid-lactation	H (1)
Property R	20	May 2018	Autumn	7 (19/280)	1 (4/280)	F, A, Holstein	Mid-lactation	H (1)
Property S	21	May 2018	Autumn	4 (30/700)	0.4 (3/700)	F, A, Holstein x Jersey	n/a	H (1)
Property T	22	September 2018	Spring	0.8 (2/250)	0 (0/250)	F, A, Holstein	n/a	B, E (2)
Property U	23	May 2019	Autumn	17 (5/30)	17 (5/30)	F, n/a, n/a	n/a	H (2)
Property V	24	April 2019	Autumn	11 (20/180)	2 (3/180)	F, A, Holstein	Mid-lactation	B, E (3)
Property U	25	June 2019	Winter	n/a	n/a	M, n/a, Angus	n/a	H (1)
Property W	26	April 2020	Autumn	28 (25/88)	3 (3/88)	F, n/a, n/a	n/a	B, E (5)

**Table 2 toxins-14-00414-t002:** Unifactorial association between weather parameters summarised over a 15-day period and the report of an acute bovine liver disease event estimated using conditional logistic regression.

Weather Parameters	Odds Ratio (95% CI)	*p*-Value
Average of daily minimum temperature (°C)	1.42 (0.98–2.05)	0.065
Average of daily maximum temperature (°C)	0.97 (0.71–1.33)	0.865
Average of daily average temperature (°C)	1.21 (0.81–1.80)	0.346
Average of daily temperature gap (°C)	1.21 (0.81–1.80)	0.346
Average of daily precipitation (mm)	1.28 (0.93–1.76)	0.134
Cumulative daily precipitation (mm)	1.02 (0.99–1.04)	0.134
Average of daily solar exposure (MJ/m^2^)	0.78 (0.56–1.10)	0.162
Cumulative daily solar exposure (MJ/m^2^)	0.99 (0.96–1.01)	0.186
Average of daily dewpoint temperature at 9 am (°C)	2.01 (1.20–3.37)	0.008
Average of daily dewpoint temperature at 3 pm (°C)	1.83 (1.15–2.90)	0.011
Average of daily minimum and dewpoint at 9 am temperature gap (°C)	0.83 (0.45–1.54)	0.557
Average of daily maximum and dewpoint at 3 pm temperature gap (°C)	0.73 (0.54–1.00)	0.047

## Data Availability

The weather dataset, weather parameter correlation matrix, Stata formatted dataset, and Stata analysis code are openly available in www.FigShare.com (accessed on 15 May 2022).

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
