# Peer review of "Investigation of Weather Triggers Preceding Outbreaks of Acute Bovine Liver Disease in Australia"

_toxins, 2022, doi:10.3390/toxins14060414_

Round 1
Reviewer 1 Report
A very interesting work, in research of triggering factors of acute bovine liver disease, a serious illness, with heavy economic losses.
The authors are aware of the limitations of the study and are well presented in the discussion. However, I have some doubts in Table 1: How many animals in each outbreak were subjected to histopathological examination is not clear. In outbreak #25 there is no indication of mortality or morbidity, and I don't understand why. The same in outbreak #4. I would like you to clarify this data.
Moreover, was the liver harvested postmortem, or in any of the live animals? Biochemistry was performed on how many animals? What do you mean by environmental exclusion for the diagnosis?
Congratulations one more time. Great work.
Author Response
Thank you for reviewing our article. I have addressed each of your suggestions below.
- I have some doubts in Table 1: How many animals in each outbreak were subjected to histopathological examination is not clear. In outbreak #25 there is no indication of mortality or morbidity, and I don't understand why. The same in outbreak #4. I would like you to clarify this data.
Response: outbreaks #4 & #25: the submitting clinician did not provide details about the number of animals at risk, affected, or dead on the submission form, so morbidity and mortality could not be calculated. I have added the following to table 1: “n/a, not available, details not provided by submitting clinician.”
- Moreover, was the liver harvested postmortem, or in any of the live animals? Biochemistry was performed on how many animals? What do you mean by environmental exclusion for the diagnosis?
Response: livers were harvested at postmortem in all cases (I have added clarification to the materials and methods: line 95. I have added the number of individuals sampled for biochemistry and/or histopathology consistent in Table 1
Kind regards,
Author
Reviewer 2 Report
This is an interesting retrospective study coming out of Australia evaluating the potential influence of climate components (i.e. dew point) within a specific geographical region on the incidence of bovine hepatic necrosis (ABLD). Overall, I found the manuscript well written, which is quite refreshing.
I do have some minor comments and suggestions. Sample size is always a challenge with retrospective field studies. Did you run a power analysis on your study to verify that your sample size met the statistical criteria.
It would be interesting to know the number of total sites in Victoria where cattle are raised. Of these, what number (%) are dairy and what number (%) are beef.
Give that your property designation (TABLE1) based on date reported coincides with the date it was reported, it is not clear if some of these properties listed on different dates are of the same sites or not. It would be helpful if that was clarified..
Table 1: Given that many of us are not familiar with the seasonal patterns/month in Australia, I would recommend that a "Season" column be added between "Outbreak Date" and "Morbidity".
Why were there no reported cases in 2009, 2012 and 2016?
Author Response
Thank you for reviewing our article. Responses are listed below.
- I do have some minor comments and suggestions. Sample size is always a challenge with retrospective field studies. Did you run a power analysis on your study to verify that your sample size met the statistical criteria.
Response: according to our study design (case cross-over study), the study sample size was dictated by the number of cases (one case window for two referent windows) accessible from the available records without sampling (case census). Therefore, the conventional pre-hoc calculation of a minimum sample size to meet a statistical criteria or expectation does not match the reality of our design. However, given potential concerns on the achieved study power suggested from marginally non-significant but of substantial amplitude estimated effect, a post-hoc effect size calculation may indeed be warranted to support continuing collection of cases in the future. Assuming a type-I-error (alpha) of 0.05 and probability of exposure of 50% among the cases, the achieved sample size of 26 cases with two matching controls provided enough study power (1-type_II-error = 80%) to detect significantly (type-I-error=5%) of 27.3% to detect an odds ratio ≥ 3.9. The following section was added to the revised manuscript at line 151: “The accessible series of 26 outbreaks, matched with two referent windows each, provided sufficient study power (1-type-II-error=80%) to detect significantly (type-I-error=5%) an odds ratio of at least 3.9 assuming a binary exposure (calculated using Stata command ‘power mcc’). This confirms that only associations of strong magnitude (OR ≥ 2 with our continuous data) could be significantly detected with the available records.”
- It would be interesting to know the number of total sites in Victoria where cattle are raised. Of these, what number (%) are dairy and what number (%) are beef.
Response: unfortunately, solid data on this isn’t available because of the property identification scheme used in Australia. I made an enquiry with the department of Agriculture in Victoria to see if they had some numbers, and they provided the following response: “There are 74,859 properties in Victoria with an active Property Identification Code (PIC) indicating they have cattle. This doesn’t mean there are 74,859 individual farms with cattle. There will be an unknown number of properties that have more than one active PIC on the same farm (some many have several). This is because some farms sell, and the new owner gets a new PIC and the previous owner forgets to advise us that they have sold/moved. Some farms have multiple PICs as they may have agistment cattle, or a sharefarmer, or operate separate enterprises on the same property. As of September 2021, there were 3,085 registered dairies in Victoria. These will be included in the 74,859. Sorry we cant be more specific on the number of cattle properties.”
I have not added this to the manuscript as these are very rough numbers, but am happy to do so if you think it is worthwhile.
- Give that your property designation (TABLE1) based on date reported coincides with the date it was reported, it is not clear if some of these properties listed on different dates are of the same sites or not. It would be helpful if that was clarified..
Response: I have added the following statement at line 90: “Some properties experienced multiple outbreaks during a similar time frame (Table 1). These were considered separate events if they commenced at distinct dates in different herds (property J), or at different geographic locations owned by the same producer (property E).” Hopefully this is acceptable clarification.
- Table 1: Given that many of us are not familiar with the seasonal patterns/month in Australia, I would recommend that a "Season" column be added between "Outbreak Date" and "Morbidity".
Response: the table has been amended according to the recommendation.
- Why were there no reported cases in 2009, 2012 and 2016?
Cases have occurred in these years, but by chance they didn’t meet the minimum inclusion criteria in our study; in such cases ABLD was highly suspected by the clinician, but histopathology wasn’t performed, and there was no written evidence that the property was thoroughly examined for other potential toxin sources.
Kind regards,
Author